# Wind Energy Harvesting with Vertically Aligned Piezoelectric Inverted Flags

**DOI:** 10.3390/s23249673

**Published:** 2023-12-07

**Authors:** Kaidong Yang, Andrea Cioncolini, Alistair Revell, Mostafa R. A. Nabawy

**Affiliations:** 1School of Engineering, The University of Manchester, Oxford Road, Manchester M13 9PL, UK; kaidong.yang@manchester.ac.uk (K.Y.); alistair.revell@manchester.ac.uk (A.R.); mostafa.ahmednabawy@manchester.ac.uk (M.R.A.N.); 2Department of Mechanical Engineering (Robotics), Guangdong Technion-Israel Institute of Technology (GTIIT), 241 Daxue Road, Shantou 515063, China

**Keywords:** wind energy, piezoelectric harvester, experiments, wall effects, interaction effects, multiple flags

## Abstract

Wind-energy-harvesting generators based on inverted flag architecture are an attractive option to replace batteries in low-power wireless electronic devices and deploy-and-forget distributed sensors. This study examines two important aspects that have been overlooked in previous research: the interaction between an inverted flag and a neighboring solid boundary and the interaction among multiple contiguous inverted flags arranged in a vertical row. Systematic tests have been carried out with metal-only ‘baseline’ flags as well as a ‘harvester’ variant, i.e., the baseline metal flag covered with PVDF (polyvinylidene difluoride) piezoelectric polymer elements. In each case, dynamic response and power generation were measured and assessed. For baseline metal flags, the same qualitative trend is observed when the flag approaches an obstacle, whether this is a wall or another flag. As the gap distance reduces, the wind speed range at which flapping occurs gradually shrinks and shifts towards lower velocities. The increased damping introduced by attaching PVDF elements to the baseline metal flags led to a considerable narrowing of the flapping wind speed range, and the wall-to-flag or flag-to-flag interaction led to a power reduction of up to one order of magnitude compared to single flags. The present findings highlight the strong dependence of the power output on the flapping frequency, which decreases when the flag approaches a wall or other flags mounted onto the same pole. Minimum flag-to-flag and flag-to-wall spacing values are suggested for practical applications to avoid power reduction in multi-flag arrangements (2-3H and 1-2H respectively, where H is flag height).

## 1. Introduction

The last few decades have witnessed numerous dramatic changes in our natural environment and a sharp rise in our global energy usage. It goes without saying that recent trends have shifted focus towards renewable and sustainable energy sources rather than conventional fossil fuels [1]. The resulting drive for sustainable energy sources further motivates interest in the harvesting or ‘scavenging’ of energy from natural resources. Ambient energy-harvesting devices are characterized by local extraction of energy on a moderate scale, offering a low-cost solution suitable for generating power in remote environments. In recent years, ambient energy harvesters have found their way to various sensing applications, including machine [2] and process [3] monitoring, air [4] and water [5] quality analysis, wildlife detection [6] and tracking [7], natural disaster prevention [8], agriculture management [9], and healthcare monitoring [10]. Ambient energy harvesting also plays a key role in distributed sensor networks and associated Internet of Things [11], which is one of the emerging fields of the so-called fourth industrial revolution. In fact, the components of a sensing node, within a wireless sensor network, typically consist of a sensor, a processing unit, a radio transceiver, and a power source, usually in the form of a battery. However, periodic battery replacement can be challenging or impractical, particularly for deploy-and-forget operations, remote environments, or large sensor networks. To overcome these limitations, self-powering technologies based on the use of ambient energy harvesters are typically considered for the previously mentioned scenarios [12], allowing for additional reductions in operational costs [13] and maintenance [14].

Among the various sources of ambient energy that can be harvested, which also include solar energy, chemical energy, and electromagnetic energy, we focus here on the harvesting of kinetic energy, which is associated with structural vibration and/or motion. Various forms of ambient kinetic energy have been successfully harvested, including vibrations caused by passing trains [15] and passing vehicles on roadways [16]; vibrations induced by human walking [17]; vibrations in automotive suspension systems [18]; vibrations or structural motion caused by fluid flows in the form of wind [19,20,21], waves [22], or water flows [23,24,25]; structural motion caused by the simultaneous action of wind and waves [26]; structural motion caused by wind plus ambient vibration [27,28]; and vibrations [29] or structural motion caused by wind [30,31] harvested simultaneously with ambient solar energy. Kinetic energy sources present in the environment show considerable variation and can be harmonic or broad-band and continuous or intermittent. Correspondingly, specialized designs of vibration energy harvesters have been proposed for harmonic excitation [32], broad-band excitation [33], and intermittent [34] or time-limited [35] excitations.

The kinetic energy harvester of interest here is the so-called inverted flag, which is specifically conceived to harvest energy from the periodic structural deformation induced by wind flow. The inverted flag is a cantilever elastic plate fixed at the downstream edge and free at the upstream edge, as shown schematically in Figure 1a. When exposed to wind flow of adequate velocity, the inverted flag exhibits large amplitude periodic oscillations, which can be exploited to harvest energy by attaching piezoelectric elements to the flag. As the flag oscillates, the piezoelectric elements convert kinetic energy into electricity. Specifically, flexible piezoelectric polymer elements, such as PVDF (polyvinylidene difluoride), are particularly suited to be incorporated into inverted flags, due to their flexibility, ease of application, acceptable energy density, robustness, and cost-effectiveness [29,36,37,38]. The response and performance of inverted flags excited by airflow have been extensively investigated within the field of piezoelectric wind energy harvesting. Typically, three dynamic response modes are known to take place in sequence for an inverted flag when the wind speed is gradually increased: (a) in-line with the incoming flow (in-line mode); (b) large amplitude limit-cycle flapping oscillation (flapping mode); and (c) deflection (deflected mode). These three response modes have also been confirmed by numerical studies [39,40,41]. The flapping mode is the most beneficial for energy harvesting purposes, as it combines the periodic flapping and the large amplitude deformation needed for the effective operation of the attached flexible piezoelectric polymers.

The feasibility of harvesting wind energy using piezoelectric inverted flags was first demonstrated by Orrego et al. [42], who conducted experiments on flags made solely from stacked PVDF strips, both indoors in a wind tunnel setup as well as outdoors in ambient wind. They showed that it was possible to use piezoelectric inverted flags to power a wireless temperature sensor operating at low power levels. Yang et al. [43] conducted a systematic experimental study to determine the impact of planform geometry on the dynamic response and energy harvesting performance of composite PVDF-based inverted flag harvesters. They confirmed the feasibility and, in fact, the superiority of a composite inverted flag; employing a core metal shim to which the piezoelectric elements are attached, rather than relying on the piezoelectric elements as the main flag structure (note that composite inverted flags are the type tested in the current work). Moreover, they evaluated the influence of aspect ratio, mass ratio, and second moment of planform area on the dynamics as well as the harvesting performance, providing results for the angular amplitude, flapping frequency, power, and power density. Later, Yang et al. [44] conducted experimental work on the durability and long operation performance of composite inverted flag harvesters. They showed that the energy produced by the harvester before breakdown due to fatigue was enough to operate a typical low-power temperature sensor for multiple months, based on a sampling rate of one measurement per minute.

Recently, the concurrent operation of two or more inverted flags has started to gain attention, with two main configurations being considered: side-by-side (see Figure 1b) [45,46,47,48,49,50,51] and tandem (see Figure 1b) [51,52,53] configurations within the horizontal plane. Huertas-Cerdeira et al. [45] experimentally studied the dynamic response of side-by-side inverted flags, with a transverse distance larger than double the length of the flag, demonstrating that the maximum angular amplitude is 36% higher than that of a single flag. Hu et al. [50] attributed this increase to the flapping promotion resulting from the leading-edge vortex (LEV) behind the adjacent flag. On the other hand, Huang et al. [52] investigated the tandem configuration and found through numerical simulations that a destructive vortex merging reduces the flapping amplitude of the downstream inverted flag by 50% compared to a single flag. However, using an experimental approach, Hu et al. [53] demonstrated that the downstream inverted flag can extract energy from the trailing edge vortex of the upstream flag at a critical flow velocity and a short streamwise distance, leading to increased angular amplitude of the downstream flag. This finding was found to be consistent with the behavior of tandem conventional flags observed by Jia and Yin [54] and Ristroph and Zhang [55].

Inverted flags were also considered in a staggered configuration along both the transverse and streamwise distances (see Figure 1b). Huang et al. [52] numerically explored two inverted flags in a staggered configuration and found that the amplitude of the downstream flag was enhanced due to constructive vortex merging. They also used simulations to investigate the influence of boundaries by changing them in the transverse direction. They found that when the gap between boundaries was small, the bending energies generated by inverted flags were much lower than isolated flags. Feng et al. [56] experimentally investigated the dynamic behavior of three inverted flags by extending the tandem configuration to either triangular or inverted triangular configurations. They reported that the dynamic response of the inverted flags with the triangular configuration was the same as that of the staggered configuration due to the weak coupling interaction between the two downstream inverted flags. On the other hand, the inverted triangular arrangement was not recommended for practical applications because the downstream inverted flag showed a smaller amplitude oscillation.

To the best of the authors’ knowledge, there has not been an in-depth study into multiple flags arranged side-by-side on the same pole, hence creating the motivation for this work. Our aim is, therefore, to study the coupling interaction between inverted flags in a vertically aligned configuration, and how it can influence the dynamic response of a flag as well as its ability to generate electrical power. We report the coupling behavior of the flags as they vertically approach each other and also when they approach a wall boundary. This is anticipated to have important implications on the realization of multi-flag harvesters where several flags are mounted onto the same pole, which is then rigidly connected to solid supports at its extremes. The rest of this paper is organized as follows: the realized flags, experimental set-up, and procedure are described in Section 2. Section 3 presents the results and insights obtained from the conducted experiments. Lastly, concluding remarks from this study are presented in Section 4.

## 2. Materials and Methods

### 2.1. Flags Tested

In this study, two sets of flags were employed: metal-only ‘baseline’ flags and metal flags with attached PVDF elements, i.e., ‘harvester’ flags. For the baseline flags, three identical flags were built, allowing for the assessment of (1) experimental repeatability (i.e., same experimental results are achieved when testing each of the three flags individually in the tunnel) and (2) interference effects (i.e., the influence of flags on each other when tested together in the tunnel). The three baseline flags, designated in this study as F1, F2, and F3, were manufactured from a stainless-steel shim resourced from Precision Brand, Downers Grove, IL, USA (www.precisionbrand.com, accessed on 1 October 2023), with density $\rho_{e}$ = 7900 kg·m^−3^, Young’s modulus $Y_{e}$ = 180 GPa, and thickness $h_{e}$ = 0.1 mm. The overhang length of the flag was 70 mm, and the width was 50 mm. As such, the flag aspect ratio is 1.4. These dimensions were selected to enable an aspect ratio slightly greater than unity, as it was shown in a previous experimental assessment that aspect ratios in such a range enable the best flapping performance [43]. Focusing on one specific aspect ratio value was deemed appropriate for a preliminary assessment of the interference effects (flag-to-wall and flag-to-flag) of interest here; future studies will generalize the observations communicated herein to other aspect ratio values, to inverted flags realized with different elastic core materials and hence with different elasticity, and to multi-flag configurations comprising more than three flags.

In addition to the baseline flags, three harvester flags were manufactured, designated F4, F5, and F6. The configuration of this variant of the flags is shown in Figure 2. The PVDF elements used were resourced from TE connectivity, Schaffhausen, Switzerland (model DT4-028K; www.te.com, accessed on 1 October 2023), with density $\rho_{p}$ = 2280 kg·m^−3^, Young’s modulus $Y_{p}$ = 2.8 GPa, piezo strain constant $d_{31}$ = 23 × 10^−12^ C.N^−1^, and thickness $h_{p}$ = 0.064 mm. Two PVDF elements/strips were bonded to each side of the core metal shim, where the core metal shim is the same as the baseline flag configuration; this way, the harvester flags F4, F5, and F6 include a core metal shim exactly analogous to the baseline flags F1, F2, and F3, and also comprise the PVDF strips and adhesive layer (see below). The original length of each PVDF strip was 171 mm, with a width of 22 mm. As such, each of the four strips employed within a flag was trimmed to a shorter length equal to that of the flag overhang length. The trimming process for each of the PVDF strips was performed using a new scalpel each time to avoid potential contamination that could result in a damaged PVDF element. The PVDF strips were bonded to the stainless-steel shim using a thin layer of double-sided adhesive tape resourced from Tesa, Hamburg, Germany (www.tesa.com, accessed on 1 October 2023), with density $\rho_{b}$ = 1100 kg·m^−3^ and thickness $h_{b}$= 0.1 mm. The two PVDF strips, employed on each side of a flag, were bonded in a way so that they were placed in a symmetric fashion around the middle line of the flag, as shown in Figure 2a. As such, only 88% of the flag width was covered with a PVDF element. While not optimal, this was deemed to be sufficient for the present study, which focuses on the effect of wall/flag proximity on power generation and not on maximizing power output. In summary, the final ‘harvester’ flag configuration is a bimorph composite inverted flag configuration, as shown in Figure 2b. It should be noted that within each PVDF strip, silver ink screen-printed electrodes covered the PVDF material from both sides, and both were wrapped in a thin mylar film.

The composite design of the present inverted flag harvesters and the associated manufacturing process are the final result of a lot of trial-and-error; based on the results that we have accumulated so far [29,30,31,37,38,43,44], this composite design is effective and robust. In fact, in the composite design, the elasticity of the flag is largely controlled by the elastic metallic core, and the inclusion of the adhesive layer and the PVDF strips adds some unavoidable damping but does not hamper the flapping capability of the flag. Basically, the elastic metallic core ‘does’ the flapping, and the PVDF strips extract energy from it. Apart from contributing to the unavoidable added damping mentioned above, the adhesive layer that keeps the PVDF strips attached to the elastic metallic core has no other noticeable effect on the flag response. This adhesive layer is very thin, tough, and proved to be very durable; in our previous study on fatigue resistance [44], the failure of the flag after half a million flapping cycles was due to a crack that originated in the elastic metallic core, while the adhesive layer and the PVDF strips did not show any sign of fatigue or structural degradation. When realizing the harvester flags used herein, we tested various prototypes and individually measured the power generated by each individual PVDF strip: the variability that we observed among different PVDF strips was always minimal at most and could be attributed to the performance variability of individual PVDF strips (as per manufacturer specs) and on the unavoidable minor imperfections due to the manual realization of the flags.

The trailing edge of the inverted flags (baseline or harvester) was rigidly fixed to a pole in the wind tunnel using custom-designed clamps that use a sandwich arrangement; see Figure 2c,d. These clamps were fabricated from laser-cut Perspex produced with a laser machine from Hobarts Laser Supplies Ltd., Rochester, UK (model AC.CLR0000.03.3020; www.hobarts.com, accessed on 1 October 2023), and allowed for a cantilever boundary condition at the flag trailing edge while ensuring a smooth outlet to the wires of the PVDF strips. During tests, the harvester flags were connected to a load resistance of 400 KΩ, corresponding to the optimum load that maximizes the power output of flag F4 when tested alone. For consistency, this value was employed for all flags during testing, as an optimum load was not identified in cases involving the operation of two or three flags.

### 2.2. Experimental Setup

Figure 3 shows a schematic of the experimental set-up used to characterize flag dynamics of the inverted flags, as well as the power generation characteristics in the case of the harvester flags. The trailing edge of the flag(s) was/were clamped and attached to a vertical pole located midway through the wind tunnel (wind tunnel manufactured by Armfield Ltd., Ringwood, UK (www.armfieldonline.com, accessed on 1 October 2023)). The wind tunnel had an octagonal cross-section with dimensions shown in Figure 3. The free-stream airflow was characterized using a calibrated hotwire anemometer before any tests were conducted, and the turbulence intensity was found to be around 0.6% within the airflow velocity range between 1.5 and 30 m·s^−1^, relevant to this study. The boundary layer thickness was within 5 mm, and the velocity profile (excluding the boundary layer) was uniform to within 1%. These conditions ensured that the tested flags were always exposed to a fully developed velocity profile during the tests. The average airflow velocity in the wind tunnel was measured using static/dynamic pressure ports connected to a calibrated pressure transducer from Sensirion AG, Stafa, Switzerland (model SDP816; www.sensirion.com, accessed on 1 October 2023) with ±5% error. The ambient conditions during testing were consistently around (298 ± 1) K and (101 ± 1) kPa.

The motion dynamics of the tested flags were recorded using a digital camera from Sony, Tokyo, Japan (model alpha 6300; www.sony.com, accessed 1 October 2023), fitted with a Sony E PZ 16–50 mm F3.5–5.6 OSS lens. Videos were captured at 100 fps with a resolution of 1920 × 1080 pixels, corresponding to a space resolution of 0.2 mm/pixel. Two black cardboard sheets were attached to the bottom and rear sides of the wind tunnel (from outside) to ensure suitable contrast for the imaging background and hence improve optical tracking accuracy. Moreover, an LED light was directed towards the transparent testing section of the tunnel for improved lighting during recording. The acquired videos were post-processed using Tracker (version 5.1.3 (https://physlets.org/tracker/, accessed 2 December 2023)), a free and open-source video analysis tool. The motion of a flag/harvester was characterized by determining the oscillation flapping frequency and angular amplitude of motion (θ in the top panel in Figure 3) at each wind speed. The angular amplitude was measured within ±5° and was defined as the angle between the two extreme flapping positions and the pole as viewed from the top. This definition gives a typical representative measure of the displacement of the flag/harvester, which has a variable curvature when deflected [30]. Note that the video recording frequency of 100 fps adopted here (which, in this present case, corresponds to about ten times the flapping frequency) was found to be adequate in identifying the amplitude of motion and the flapping frequency of interest, and follows on from our previous work [43] where we used the same experimental setup to analyze inverted flags flapping with frequencies in the range of 2–10 Hz. Measurements were repeated several times with increasing and decreasing wind flow velocity, demonstrating good repeatability within the mentioned measuring errors.

The power output from the harvester flags was collected using a National Instruments, Austin, TX, USA (NI) model NI-USB-6225 external DAQ device (www.ni.com, accessed 2 December 2023). Data were processed using LabVIEW 2017, where the data acquisition program was written as a Virtual Instrument (VI) using the DAQ-mx library (version 20.1.0.0). The program collected, saved, and displayed real-time data for instant power output observation. The data were sampled at a rate of 1 kHz to provide sufficient resolution during a flapping cycle.

### 2.3. Conducted Tests

The adopted testing sequence started at a low wind speed, with the flag(s) in an in-line position (i.e., aligned with the incoming flow). Wind speed was then increased in a stepwise manner first through the flapping mode (i.e., large flapping oscillation) and then through the deflected mode (i.e., the flag was deflected to one side). The sequence was then completed in reverse, i.e., the wind speed was gradually and stepwise decreased until the flag(s) resumed their original state (in-line mode). By doing this, we could make sure that all flag(s) had the same initial state for both increasing and decreasing speed measurements. Conducting the tests in this manner also permitted an evaluation of the level of hysteresis present, which was found to be negligible. As such, in the following section, we present only results from the decreasing flow experiments to avoid overcrowding.

We first started by testing the typical configuration, with minimal interference, whereby a single inverted flag was fixed at the middle of the tunnel. We repeated this test for the three baseline flags to ensure they all provided an identical performance, i.e., no noticeable differences due to manufacturing imperfections or setup constraints. Once that test was established, we conducted measurements on both baseline and harvester flags to assess their performance when they approached a boundary, whether that be a wall or another flag. As such, three sets of configurations were explored, as schematically presented in Figure 4. We first considered measuring one flag approaching the tunnel wall, while varying the gap distance between the flag and the wall. Clearly, this configuration leads to a non-symmetric setup where the flag is close to a boundary from one side and free from any interactions from the other side. Next, we employed two flags, allocating them symmetrically in the middle of the tunnel. Here, the only variable changed is the gap distance between the two flags. As such, this configuration allows for assessing the change of performance of a flag when approached by a moving boundary (in this case another flag) from one side. Lastly, we employed three flags where an intermediate flag is approached from both sides by two other similar flags in a symmetric fashion. Again, the only variable in this setup is the gap distance between any two flags. For each configuration in Figure 4, measurements were carried out for different values of the distance d. Given the focus of this study, these values were identified empirically during the tests to best demonstrate the detrimental effect of approaching the boundary on the flag dynamic response; i.e., as opposed to testing for a predefined range of the parameter d.

Wind tunnel blockage effects need to be accounted for carefully, particularly as the number of flags increases. As shown in Figure 3, the tunnel employed in this study has an octagonal cross-section with an area of 87,823 mm^2^. The projected area of the flag(s) will vary depending on the number of flags installed in the tunnel and the dynamics experienced. The projected area was observed to have an upper bound of 80% of the lateral area, based on the observations made in our work. This can be conservatively evaluated as 0.8 × 50 mm × 70 mm = 2800 mm^2^. Consequently, the corresponding values for the blockage ratio for the one-, two-, and three-flag configurations tested were 3.2%, 6.4%, and 9.6%, respectively. It is generally understood that blockage ratios of the order of 5–10% are inconsequential. Indeed, recent research indicates that blockage effects on drag measurements are negligible up to blockage ratio values of 15–16% for both streamlined and blunt rigid profiles [57], while wind-induced vibrations seem less affected by blockage than drag [58]. As such, it is a reasonable assumption that blockage effects in the present setup can be neglected.

## 3. Results and Discussion

### 3.1. Overview

For clarity, this section is split into two parts: results for baseline flags (F1, F2, and F3) are presented first, followed by the results for the harvester flags (F4, F5, and F6). We restrict our attention to the flapping regime only, as this is the operation mode of interest for the energy harvesting application.

### 3.2. Baseline Flags

Preliminary measurements for the baseline flags are provided in Figure 5, where the amplitude and frequency of flapping are presented as a function of the wind speed. In these preliminary tests, each flag was tested alone in the middle of the wind tunnel. Being that these flags were nominally identical, the recorded dynamics should be the same as within measuring errors. As can be noted in Figure 5, this is indeed the case, hence confirming that any manufacturing imperfections are inconsequential.

The next set of experiments, where flag F1 was tested at different distances from the wall of the wind tunnel (first configuration in Figure 4), are documented in Figure 6. As the flag approached the wall, the wind speed range of flapping gradually shrank and shifted towards lower velocities, from 12.2 to 25.2 m·s^−1^ when the flag was placed in the middle of the wind tunnel, to 10.8 to 22.7 m·s^−1^ when the flag was placed at d = 30 mm from the wind tunnel wall. Moreover, it is evident that, for a given wind speed, the flapping amplitude generally increases due to wall proximity, whereas the flapping frequency decreases in proportion to the wind speed; the higher the windspeed, the more pronounced the reduction.

The results of the experiments where flag F1 was tested while approached by other flags in the second and third configurations of Figure 4 are provided in Figure 7 and Figure 8. It is evident that as the flag was approached by another flag from one side, the wind speed range of flapping gradually shrank and shifted towards lower velocities from 12.2 to 25.2 m·s^−1^ when the flag was alone in the middle of the wind tunnel to 11.5 to 22.7 m·s^−1^ when the separation distance between the two flags was just 17 mm. This effect is even more pronounced when the flag is approached from both sides, as the wind speed range of flapping was reduced to 11.1–22.3 m·s^−1^ when the separation distance between the three flags was 17 mm.

The flapping frequency always decreases in proportion to the wind speed and distance from the other flag or flags; the more pronounced the reduction, the higher the wind speed, the lower the separation distance, which is the same qualitative trend observed when the flag approaches the wind tunnel wall. On the other hand, the flapping amplitude can increase, decrease, or remain unaffected, depending on the wind speed and flag separation distance.

### 3.3. Harvester Flags

The results of the experiments where the harvester flag F4 was tested at different distances from the wall of the wind tunnel (first configuration in Figure 4) are documented in Figure 9. Note that the main objective of this present study was to assess the relative power reduction observed with flag F4 due to the proximity effects, and not to ascertain the maximum power that this harvester could achieve (which would have required the fine-tuning of the load resistance during each test). Power measurements are correspondingly reported in the subsequent figures with normalized values, which is the most effective way to assess relative variations. For a more detailed investigation of the absolute power output of this flag construction for a range of different geometries, the reader is referred to our previous contribution [43]. The first thing to observe from inspecting Figure 9 is that the wind speed range of flapping of F4 alone while in the middle of the tunnel was considerably narrower in comparison to its corresponding baseline flag F1 (compare the blue and red trends in Figure 9). This is unsurprising since the inclusion of the PVDF strips acts to increase the damping of the flag, while the flexural rigidity (determined by the metal shim) remains largely unaffected, as reported previously in Ref. [30]. As such, the observed reduction of the wind speed range of flapping can be attributed to the increase in damping. Remarkably, even though the wall proximity only causes a minor reduction in the flapping frequency of flag F4 (Figure 9b), while the flapping amplitude is practically unaffected (Figure 9a), the reduction in power (Figure 9c) is rather pronounced and reaches about 25% at the highest wind speed and separation distance of 50 mm.

As demonstrated in Ref. [59], the power output of any motion-driven energy harvester scales, as a first approximation, in proportion to the product of the travel range of the moving mass times the cube of the oscillation frequency, which, in this present case, corresponds to the product of the flapping amplitude times the cube of the flapping frequency. As can be noted in Figure 9c,d, the trends of the RMS power and those of the product of the flapping amplitude times the cube of the flapping frequency compare fairly well (remember that the mentioned proportionality between the power and the product of amplitude times the cube of the frequency is only a first approximation; several factors are not taken into account, notably the structural damping and any added damping resulting from proximity effects). Therefore, the minor reduction in flapping frequency, combined with the strong dependence of the power output on frequency, can be regarded as the leading cause of power degradation from wall proximity.

The results of the experiments where the harvester flag F4 was tested while being approached by other flags (i.e., second and third configurations in Figure 4) are provided in Figure 10 and Figure 11. It is evident that the narrowing of the flapping wind speed range is more pronounced the closer the flags and the frequency of flapping is also substantially reduced by the proximity effects. The reduction in power is correspondingly quite pronounced, up to one order of magnitude in comparison with the flag tested alone. Inspection of Figure 10c,d and Figure 11c,d clearly shows that the reduction in power is now more pronounced than what would be expected on the basis of the reduction in flapping amplitude and frequency, hence indicating that the flag-to-flag interference is more detrimental to power production than the flag-to-wall effects. Even though not explicitly investigated at this stage and left for future developments, an increase in damping due to the coupling among different flags, and/or a modification of the flag curvature distribution during flapping, might be responsible of the observed power degradation, in addition to the reduction in flapping amplitude and frequency.

## 4. Concluding Remarks

We conducted a systematic experimental study into the effects of interactions between multiple inverted flags that were vertically aligned, a configuration not yet explored in the literature. Three identical baseline flags and the corresponding harvester flags with attached PVDF elements were fabricated and tested in different vertical alignment configurations, including one flag approaching the wind tunnel wall and two/three flags approaching each other in the middle of the wind tunnel. Measurements included the flapping amplitude, flapping frequency, and root–mean–square power (for harvester flags).

We found that approaching a wall, or the inclusion of more flags in a vertical alignment configuration, generally have detrimental effects on the dynamic response and power generation performance compared to operating a typical single flag; the flapping speed range shrinks, the flapping frequency decreases, and the power output significantly drops (as a consequence of the strong dependence of the power output on flapping frequency). Based on the present results, when multiple flags are located on the same pole, the power output reduction caused by proximity effects can be avoided if the distance between adjacent flags is on the order of two to three times the flag height, and the distance between the flags and solid walls is on the order of one to two times the flag height.

Clearly, further future work is needed to consolidate these findings for a wider range of geometries (aspect ratio and planform shapes), mass distributions, and bending stiffnesses. In order to achieve a better fundamental physical understanding, future experiments should ideally also include the time-resolved measurement of the flow field that develops around the inverted flags, and/or the development of dedicated fluid–structure-interaction numerical methodologies, in order to complement the structural dynamics measurements and help interpret the observations. That said, we believe our findings pave the way for more focused studies on this configuration and are insightful for researchers working on the realization of multi-flag energy harvesting systems.

## Figures and Tables

**Figure 1 sensors-23-09673-f001:**
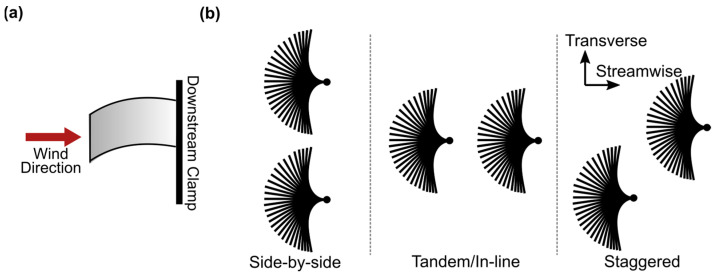
(**a**) Schematic of the inverted flag configuration. (**b**) Schematics of the different multiple inverted flag configurations considered in the literature (shown from a top view for clarity)**,** including side-by-side; tandem/in-line; and staggered configurations (wind flow is left-to-right).

**Figure 2 sensors-23-09673-f002:**
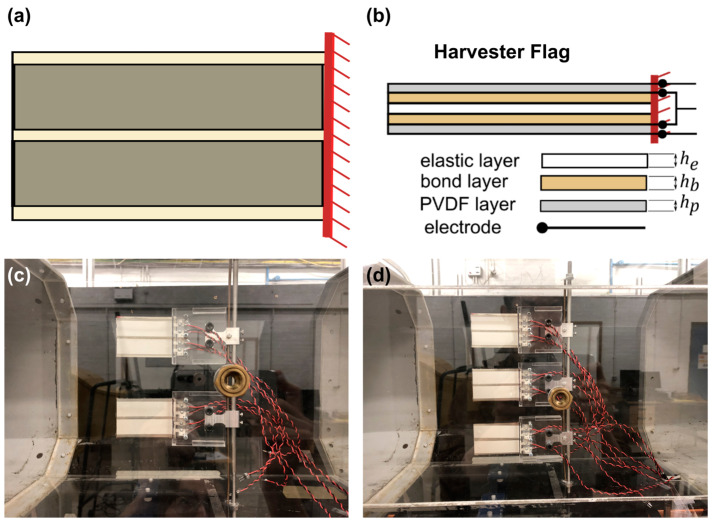
(**a**) Side view schematic of the harvester flag configuration (the fixed pole of the harvester flag is located on the right and is vertically oriented, and the wind flow is left-to-right); (**b**) top view cross-sectional schematic of the bimorph composite inverted flag construction (note that the PVDF layer does not fully cover the flag); (**c**,**d**) two and three vertically aligned flags as tested in the wind tunnel.

**Figure 3 sensors-23-09673-f003:**
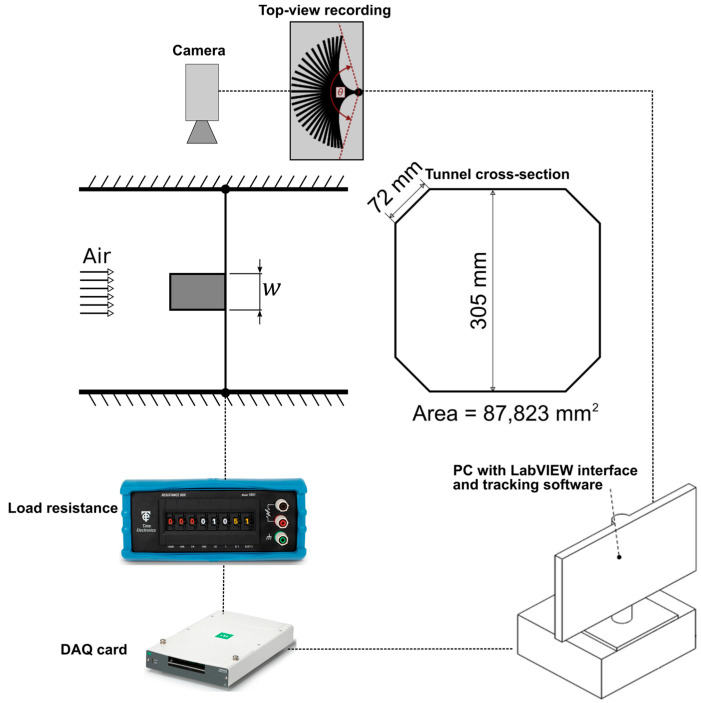
Schematic of the experimental set-up used to characterize the dynamics and the power generation of the flag(s) (note that the flapping angular amplitude θ used to characterize the flag motion is defined in the top panel).

**Figure 4 sensors-23-09673-f004:**
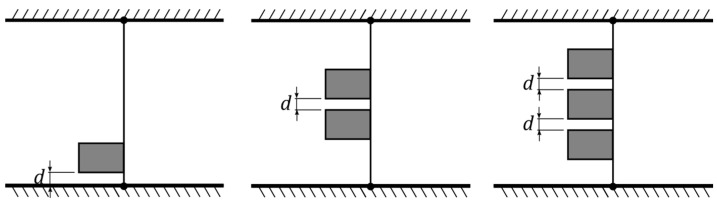
Schematic representation of the different flag configurations explored in this study.

**Figure 5 sensors-23-09673-f005:**
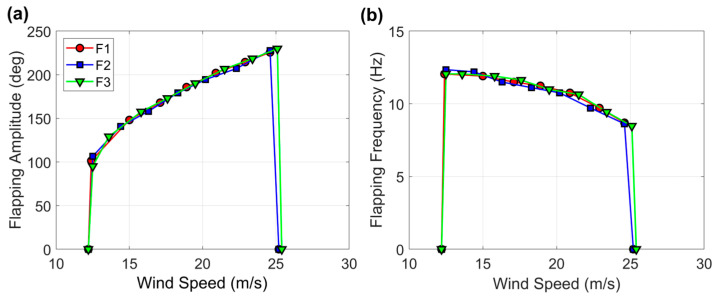
(**a**) Flapping amplitude and (**b**) flapping frequency for baseline flags F1, F2, and F3 in typical configuration, as shown in Figure 3.

**Figure 6 sensors-23-09673-f006:**
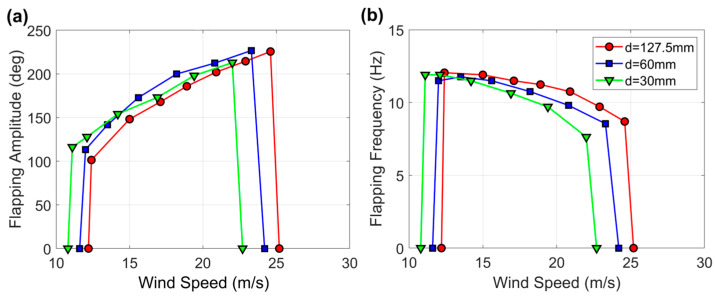
(**a**) Flapping amplitude and (**b**) flapping frequency for the baseline flag F1 while approaching the tunnel wall (first configuration in Figure 4). Note that d = 127.5 mm corresponds to flag F1 placed in the middle of the wind tunnel.

**Figure 7 sensors-23-09673-f007:**
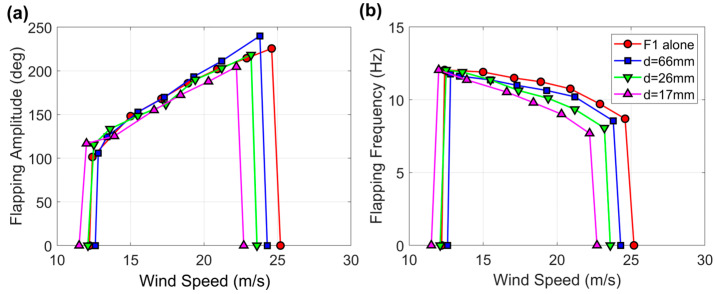
(**a**) Flapping amplitude and (**b**) flapping frequency for the baseline flag F1 while being approached by another flag (second configuration in Figure 4). The measurements for flag F1 alone in the middle of the wind tunnel are also included for comparison.

**Figure 8 sensors-23-09673-f008:**
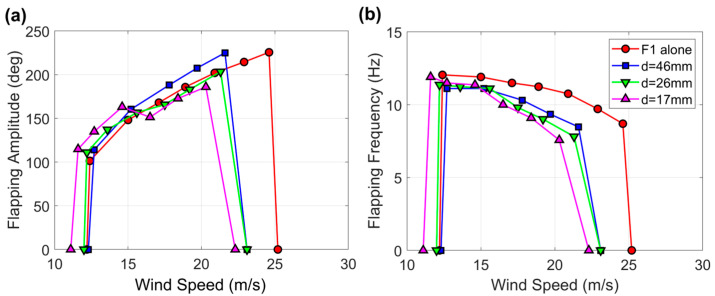
(**a**) Flapping amplitude and (**b**) flapping frequency for the baseline flag F1 while being approached by two other flags (third configuration in Figure 4). The measurements for flag F1 alone in the middle of the wind tunnel are also included for comparison.

**Figure 9 sensors-23-09673-f009:**
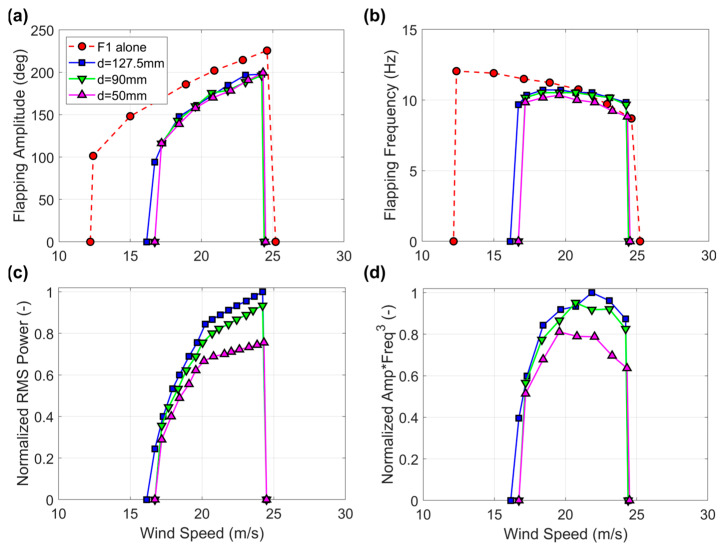
(**a**) Flapping amplitude, (**b**) flapping frequency, (**c**) normalized RMS power, and (**d**) normalized product of flapping amplitude times the cube of the flapping frequency for the harvester flag F4 while approaching the wind tunnel wall (first configuration in Figure 4). Note that d = 127.5 mm corresponds to flag F4 placed in the middle of the wind tunnel. Baseline flag F1 included for comparison in panels (**a**,**b**).

**Figure 10 sensors-23-09673-f010:**
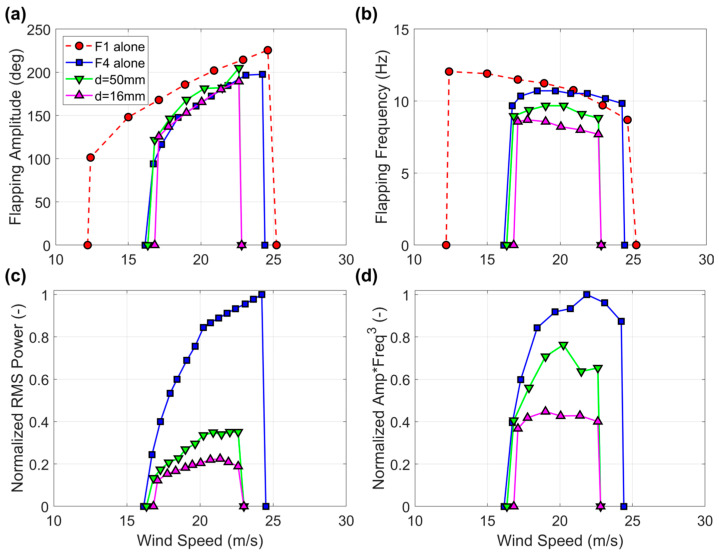
(**a**) Flapping amplitude, (**b**) flapping frequency, (**c**) normalized RMS power, and (**d**) normalized product of flapping amplitude times the cube of the flapping frequency for harvester flag F4 while being approached by another flag (second configuration in Figure 4). Baseline flag F1 included for comparison in panels (**a**,**b**).

**Figure 11 sensors-23-09673-f011:**
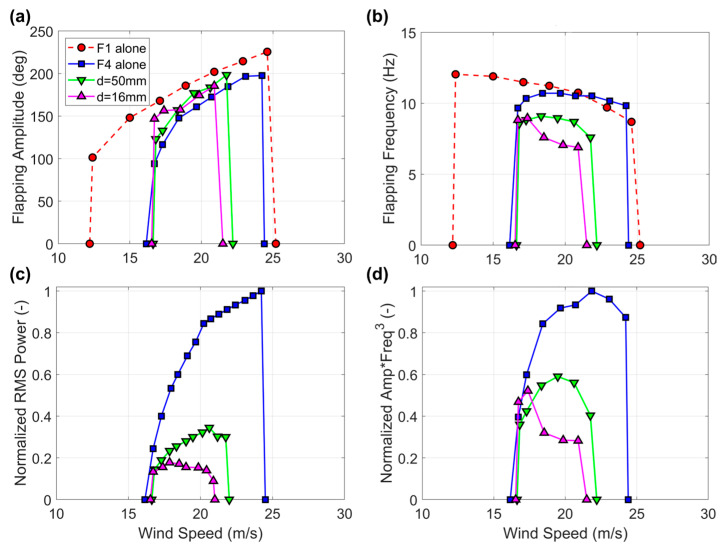
(**a**) Flapping amplitude, (**b**) flapping frequency, (**c**) normalized RMS power, and (**d**) normalized product of flapping amplitude times the cube of the flapping frequency for harvester flag F4 while being approached by two other flags (third configuration in Figure 4). Baseline flag F1 included for comparison in panels (**a**,**b**).

## Data Availability

All data presented are contained within the article.

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
