# Peer review of "Wind Energy Harvesting with Vertically Aligned Piezoelectric Inverted Flags"

_sensors, 2023, doi:10.3390/s23249673_

Round 1

Reviewer 1 Report

Comments and Suggestions for Authors

In order to obtain a more accurate comparison, it is imperative to ensure that the mechanical and electrical properties of each piezoelectric beam are entirely consistent under identical conditions, thereby eliminating the possibility that these differences are attributed to the inherent characteristics of the piezoelectric beam. In my perspective, the mechanical and electrical characteristics of the piezoelectric beam are significantly influenced by the adhesive layer, and it is challenging to ensure uniformity in these aspects among different piezoelectric beams. Thus, I am interested in understanding the approach undertaken by the author to mitigate this interference.

Comments on the Quality of English Language

fine

Reviewer 2 Report

Comments and Suggestions for Authors

This paper presented a multiple inverted flag arrangement for the better amplitude/frequency response. The experiment is conducted well but the results do not compare with theoretical or simulation to obtain the accuracy. The video is used 100 fps to capture the frequency response of around 12 Hz. The reviewer think that the fps is not high enough to reflect the accuracy.

In addition, six different flag F1 to F6 was reported in two groups. The first group F1, F2 and F3 were manufactured from a stainless-steel and the second group from PVDF (polyvinylidene difluoride) piezoelectric polymer elements. It will be crucial if the author could highlight extensive differences between each respective flags F1, F2 and F3 designs as no clear distinction on geometry was mentioned in reference to performance preference in Fig. 5.

More in-depth analysis should be performed to reflect the overall analysis. Fig 9(c), presented the normalized power, it could not identify exactly the power and bandwidth. The authors should report the power only (without normalized it).

Comments on the Quality of English Language

The quality of the English language is good.

Reviewer 3 Report

Comments and Suggestions for Authors

This paper presents an experimental study on the use of multiple inverted flags to scavenge energy from wind flow.

The paper is interesting and well written.

The following points should be addressed to improve the overall quality of the paper.

1.

The title is not well representative. It should be highlighted the application (i.e. wind) and the type of harvester (i.e. kinetic or vibration).

 2.

In the introduction, a reference should be added where the first occurrence of “inverted flag” is used.

 3.

As it is stated, it seems that refs [2]-[10] exploit the "inverted flag config.". This should be clearly stated as not to confuse the reader.

 4.

Also, at the stage it is firstly mentioned, it is not clear if the "inverted flag" is a sort of configuration to scavenge energy from kinetic, thermal or solar. These different types of energy sources are mentioned later on, and this generates confusion.

 5.

The references to general harvesters can be out of focus. In particular, references to thermal energy, solar energy, electromagnetic energy, and chemical energy detract the focus from the one of interest, which is from kinetic energy and induced vibrations.

I would recommend focusing on kinetic (or vibration) energy harvesters only, and expand the literature review to other applications, e.g., energy harvesting from passing vehicle induced vibration and walking persons. These are, for example, discussed in:

 - Energy harvesting from the vibrations of a passing train: Effect of speed variability, Journal of Physics: Conference Series 744(1),012080, 2016

- “Piezoelectric energy harvesting from human walking using a twostage amplification mechanism”, Energy 189 (2019) 116140.

 6.

Also, different types of vibrating sources are available in the environment, which span from sinusoidal, random, intermitted or time-limited excitations. These are treated respectively, for example, in:

 - Mono-stable and bi-stable magnetic spring based vibration energy harvesting systems subject to harmonic excitation: Dynamic modeling and experimental verification, Mechanical Systems and Signal Processing, 134, 106361, 2019

- Vibration energy harvesting from random force and motion excitations, Smart Materials and Structures, 21(7), 075025, 2012

- Challenges for Energy Harvesting Systems Under Intermittent Excitation, IEEE J. Emerging Sel. Top. Circuits Syst., 4(3), pp. 364–374, 2014

- Harvesting Energy From Time-Limited Harmonic Vibrations: Mechanical Considerations, Journal of Vibration and Acoustics, 139, 051019, 2017

 I believe that the interested reader should be made aware that there are a series of different input from which to harvest kinetic energy.

 7.

Although ref [51] is mentioned to motivate the aspect ratio of the flags, how were the dimensional values of the parameters selected? is there any design process, or application? Why not length 100 mm and height 70 or any other combination?

 8.

How is the "flapping amplitude (deg)" defined?

Is it identified by the angle between the two red-dashed lines in fig 3? it should be clearly defined and identified.

 9.

How is the power output measured?

 10.

The paper only reports the results of the experimental study.

An effort should be devoted to deeply interpret the results shown, giving them a physical interpretation and trying to identify a relation among the flapping amplitude, frequency and power output.

Since there is no modelling section, the paper should not be limited to show results, but giving them a better understanding to the reader.

11.

Is there any model in the literature that can be used for validation?

It would be interesting to compare the experimental results to a simple model even using a single inverted flag.

Round 2

Reviewer 3 Report

Comments and Suggestions for Authors

The authors have addressed the reviewer's comments and the paper has been improved